# Effect of antithrombotic stewardship on the efficacy and safety of antithrombotic therapy during and after hospitalization

Albert R. Dreijer [1,2]*, Marieke J. H. A. Kruip[3,4], Jeroen Diepstraten[2], Suzanne Polinder[5], Rolf Brouwer[6], Peter G. M. Mol[7], F. Nanne Croles[3], Esther Kragten[6], Frank W. G. Leebeek[3], Patricia M. L. A. van den Bemt[7]

1 Department of Hospital Pharmacy, Erasmus University Medical Center, Rotterdam, The Netherlands, 2 Department of Hospital Pharmacy, Reinier de Graaf Hospital, Delft, The Netherlands, 3 Department of Hematology, Erasmus University Medical Center, Rotterdam, The Netherlands, 4 Thrombosis Service STAR-SHL, Rotterdam, The Netherlands, 5 Department of Public Health, Erasmus University Medical Center, Rotterdam, The Netherlands, 6 Department of Hematology, Reinier de Graaf Hospital, Delft, The Netherlands, 7 Department of Clinical Pharmacology, University of Groningen, University Medical Center Groningen, Groningen, The Netherlands

* a.dreijer@treant.nl

**Data Availability Statement:** All relevant data are within the paper and its Supporting Information files.

## Abstract

### Background

Although the benefits of antithrombotic drugs are indisputable to reduce thrombotic events, they carry a high risk of compromising patient safety. No previous studies investigated the implementation and (cost-) effectiveness of a hospital-based multidisciplinary antithrombotic team on bleeding and thrombotic outcomes. The primary aim of this study was to compare the proportion of patients with a composite end point consisting of one or more bleeding episodes or one or more thrombotic event from hospitalization until three months after hospitalization.

### Methods and findings

A prospective, multicenter before-after intervention study was conducted in two Dutch hospitals. Adult patients hospitalized between October 2015 and December 2017 treated with anticoagulant therapy were included. The primary aim was to estimate the proportion of patients with a composite end point consisting of one or more bleeding episodes or one or more thrombotic event from hospitalization until three months after hospitalization. The intervention was the implementation of a multidisciplinary antithrombotic team focusing on education, medication reviews by pharmacists, implementing of local anticoagulant therapy guidelines based on national guidelines, patient counselling and medication reconciliation at admission and discharge. The primary endpoint was analysed using segmented linear regression. We obtained data for 1,886 patients: 941 patients were included in the usual care period and 945 patients in the intervention period. The S-team study showed that implementation of a multidisciplinary antithrombotic team over time significantly reduced the composite end point consisting of one or more bleeding episodes or one or more thrombotic

**Funding:** Stichting Phoenix Schiedam, the pharmaceutical companies (Daiichi Sankyo, Boehringer Ingelheim, Bayer and Pfizer) and the Scientific Committee Reinier de Graaf Gasthuis provided financial support for this study in the form of unrestricted grants. The funders had no role in study design, data collection and analysis, decision to publish, or preparation of the manuscript.

**Competing interests:** I have read the journal's policy and the authors of this manuscript have the following competing interests: FWGL received unrestricted research grants for studies on Von Willebrand disease. He is consultant for uniQure, Takeda and NovoNordisk, of which fees go to the university. He is a DSMB member for a study of Roche. All these potential COI are not related to the current study and does not alter our adherence to PLOS ONE policies on sharing data and materials . ARD, MJHAK, JD, SZ, RB, PGMM, FNC, EK, and PMLAvdB have no conflict of interest.

event from hospitalization until three months after hospitalization in patients using anticoagulant drugs (-1.83% (-2.58% to -1.08%) per 2 month period).

## Conclusions

This study shows that implementation of a multidisciplinary antithrombotic team over time significantly reduces the composite end point consisting of one or more bleeding episodes or one or more thrombotic event from hospitalization until three months after hospitalization in patients using anticoagulant drugs.

## Trial registration

Trialregister.nl NTR4887.

## Introduction

Although the benefits of antithrombotic drugs are indisputable to reduce thrombotic events, they carry a high risk of compromising patient safety in terms of bleeding [1–4]. Several studies have suggested that antithrombotic management initiatives improve patient outcomes. Bajorek et al. implemented a pharmacist-coordinated multidisciplinary review process in a hospital setting to optimize antithrombotic use in elderly with atrial fibrillation (AF). As a result of the intervention, 35.8% (78 out of 218) of the patients required adaptation of their existing antithrombotic therapy [5]. Schillig and colleagues showed that implementation of an inpatient pharmacist-directed anticoagulation service focusing on transition of care from the inpatient-to-outpatient setting led to improvement in patient handoff, improved communication, and earlier patient follow-up after discharge. However, no impact on bleeding and thrombotic outcomes was observed [6]. Padron and Miyares described an expanded antithrombotic stewardship focusing on medication surveillance for inpatients on antithrombotic therapy. Outcomes measured were protocol adherence for dabigatran, heparin, argatroban and attainment of therapeutic levels for heparin infusions. By implementation of stewardship, the length of hospital stay was reduced by 1.5 days and cost-savings were $661 per patient over 1.5 years based on 409 patients on anticoagulants [7]. Most studies focused on patients treated with warfarin [6] for specific indications, such as AF or venous thromboembolism (VTE) [5,8]. As far as we know, no previous studies investigated the implementation and (cost-) effectiveness of a hospital-based multidisciplinary antithrombotic team on bleeding and thrombotic outcomes. Therefore, we designed the S-team study (antithrombotic stewardship study; in Dutch: Stollingsteam), to study the effect of implementation of a hospital-based multidisciplinary antithrombotic team on the efficacy and safety of antithrombotic therapy during and after hospitalization. The team focused on education, medication reviews by pharmacists, implementing of local anticoagulant therapy guidelines based on national guidelines, patient counselling and medication reconciliation at admission and discharge [9].

## Methods

### Study design and setting

The S-team study was a prospective before-after intervention study with an interrupted time series design performed in the Erasmus University Medical Center (EMC, 1320 beds) and a

large general teaching hospital (Reinier de Graaf Hospital; RdGG, 590 beds). A comprehensive paper on the S-team study protocol has previously been published [9].

## Study population

Patients aged 18 years and older who were admitted to the EMC or RdGG between October 2015 and December 2017 and treated with therapeutic anticoagulant medication were eligible for inclusion. The study population consisted of patients who started with anticoagulant therapy in the hospital, patients who were already treated with anticoagulant therapy before hospitalization and patients who restarted anticoagulant therapy after a surgical or non-surgical intervention. Due to the limited availability of study personnel, we maximised recruitment to three patients per day per hospital. A random number generator was used to select those three patients. Only the patient's first hospital admission was included. All participants provided written informed consent during hospitalization. Exclusion criteria were: (1) no informed consent from the patient, (2) hospitalization for less than 24 hours, (3) admission to the intensive care unit (ICU) without (previous or subsequent) admission to a general care ward, (4) patients treated with low-molecular-weight heparins (LMHWs) only in a prophylactic dose, (5) patients treated with a single dose of an anticoagulant (e.g. heparin flush).

Thrombotic and bleeding complications, length of hospitalization, all-cause mortality and medical costs were compared between a 12-month usual care period (pre-intervention) and a 12-month intervention period. The intervention was the implementation of a hospital-based multidisciplinary antithrombotic team. Because the study did not fall under the scope of the Medical Research Involving Human Subjects Act, a waiver was obtained from the Medical Ethics Committee of the Erasmus University Medical Center (MEC-2015-386). The S-team study was registered in the Netherlands Trials Registry, number NTR4887 at www.trialregister.nl.

## Usual care

During the usual care period (October 2015 to September 2016) the normal procedures of medication surveillance, patient counselling and medication reconciliation at admission and discharge were maintained. A detailed description of the procedures during the usual care period can be found in our previously published study protocol [9].

## Intervention

The intervention consisted of the implementation of a multidisciplinary antithrombotic team. In both hospitals the team consisted of a specialized thrombosis nurse as case manager, a haematologist, a hospital pharmacist/clinical pharmacologist, a cardiologist, an anaesthesiologist, a pulmonologist, a paediatrician, a neurologist and a surgeon. In the university medical center a haematologist, who is also (head) of the regional thrombosis service and a quality officer belonged to the team and in the general hospital, a clinical chemist and an emergency physician were part of the team. The teams focused on the following interventions:

**Education.** To increase the knowledge of antithrombotic therapy among physicians, nurses and hospital pharmacists, hospital-wide education was given.

**Medication reviews by pharmacists.** Daily structured medication reviews were performed by the pharmacist focused on optimizing treatment with all types of anticoagulants. The pharmacotherapy review comprised of checks on dosing (i.e., in relation to decreased renal function, bodyweight and age), duplicate medication (in specific double and triple antithrombotic treatment), drug–drug interactions, contraindications and perioperative bridging of anticoagulant treatment during surgery or interventions.

**Antithrombotic therapy guidelines.** Local guidelines were drafted based on recent national guidelines and updated to ensure there was a uniform policy on antithrombotic therapy in both hospitals [10].

**Patient counselling.** The purpose of patient counselling was to provide information and education to patients with the aim of giving the patient more control and responsibility on their own health, with a specific focus on antithrombotic therapy. This patient empowerment was performed on a daily basis for each included patient.

**Medication reconciliation.** At admission, recent data from the patient's thrombosis service regarding vitamin K antagonist (VKA) therapy were handed over to the responsible physician. The data consisted of the dosing scheme, indication for VKA therapy, type of VKA, INR measurements and the INR target range. At discharge, pharmacotherapy advices regarding all types of anticoagulants (i.e. VKAs, direct oral anticoagulants (DOACs) and LMWHs) from the medication reviews were handed over to the thrombosis service (in case of VKAs), the general practitioner and to the community pharmacist.

## Data collection

Data on the clinical outcomes and healthcare use were collected from electronic patient records in the hospital information systems (HiX; Chipsoft, Amsterdam, the Netherlands and Elpado; home-built system Erasmus University Medical Center, Rotterdam, the Netherlands) (S1 Table). Data were collected from the day of hospitalization or time of starting the first anticoagulant therapy or from the day of discharge from the ICU to a general care ward until three months after hospitalization or patient death. For patients who were initially admitted to a general care ward and subsequently transferred to the ICU, data were collected from the day of hospitalization until admission to the ICU. Costs were calculated from a healthcare perspective including hospital medical costs. Costs (labour costs S-team, costs for bleeding/thrombotic events and costs for hospital days, including ICU days) were calculated for both the usual care and intervention period for each hospital separately. Labour costs were calculated by multiplying the time spent on the activities (S-team meetings, medication reviews by pharmacists, patient counselling, drafting and maintenance of anticoagulant therapy protocols and education) by salary expenditures of healthcare providers. The number of bleeding events and the number of thrombotic events were calculated and multiplied by the costs of the specific event. Data from earlier studies were used to define costs per bleeding or thrombotic event. Cost of a major bleeding was €5,949 (£5,067; $6,739) and cost of a non-major bleeding was €4,378 (£3,729; $4,960) [11]. Costs of thrombotic events were divided into arterial thrombosis (€4,790 (£4,077; $5,423)) [12], deep vein thrombosis (€4,449 (£3,789; $5,040)) and pulmonary embolism (€7,736 (£6,589; $8,764)) [13]. The difference in costs for hospitalization days before and after implementation of the S-team was calculated by multiplying the mean of hospitalization days by the costs of one hospitalization day (€642 (£546; $726)) for the University Medical Center and €443 (£377; $501) for the general hospital [14]. All data were processed with Open Clinica (Open Clinica LLC, Waltham, USA).

## Outcome

Primary outcome was the proportion of patients with a composite end point consisting of one or more bleeding episodes or one or more thrombotic event from hospitalization until three months after hospitalization. The three month follow-up period was justified by the assumption that the period shortly after hospitalization represents a period of instability for the patient. Patients with bleeding or thrombotic events as a reason for admission were also eligible for inclusion; however, these events leading to the hospital admission were not included in the primary endpoint.

Bleeding was defined as a composite endpoint of major bleeding and non-major bleeding according to the International Society on Thrombosis and Haemostasis (ISTH) criteria [15,16]. A thrombotic event was defined as any objectively confirmed arterial or venous thrombosis, including acute myocardial infarction or ischemic stroke for arterial thrombosis and deep venous thrombosis or pulmonary embolism or venous thrombosis and any objectively determined arterial or venous thrombus at other sites [17–20]. The bleeding and thrombotic events were evaluated and classified according to the ISTH criteria by two independent expert physicians in the field (FNC and EK). All case record forms were blinded with respect to the study period. Discrepancies between the assessments of the expert physicians were discussed to reach final consensus.

Secondary outcome was the proportion of patients with a major and non-major bleeding event and the proportion of patients with a fatal and non-fatal thrombotic event. Additional secondary outcomes were the proportion of patients with a composite end point consisting of one or more bleeding episodes or one or more thrombotic event during hospitalization, the proportion of patients with a composite end point consisting of one or more bleeding or one or more thrombotic event after hospitalization, all-cause mortality, length of hospitalization, and medical costs.

## Sample size

On the basis of the available literature, we estimated that the proportion of patients with a composite end point consisting of one or more bleeding or one or more thrombotic event would be 9% [21–23]. Our study was powered to decrease this to a composite rate of 5.5%. A Chi Square test with a type 1 error of 0.05, power 80% resulted in a sample size of 1,834 patients. In order to account for drop-outs, we aimed to include 1900 patients.

## Data analysis

All data were analysed with IBM SPSS Statistics version 21.0 (IBM Software, New York, USA). All continuous variables were tested for normality with the Shapiro-Wilk test. Non-normal variables were expressed as medians and interquartile ranges (IQR) and differences between groups tested with the Mann-Whitney U test. Normal variables were expressed as mean and standard deviation (SD) and difference between groups tested with the t-test. Categorical variables were presented as percentages and tested for statistical significance between groups using the Chi square test. $P < 0.05$ was considered to be statistically significant.

A predefined statistical plan, as stated in our previously published study protocol was used for analysis [9]. For analysis of the primary outcome we used segmented regression analysis for the interrupted time series (ITS) data. The data points for the time series data represent the proportion of patients with a bleeding or thrombotic event aggregated by inclusion date per two months (i.e., six data points before and six data points after the intervention each consisting of at least 30 patients). The interruption was the implementation of the multidisciplinary antithrombotic team (October 2016 to December 2016). Durbin-Watson statistics was used to check for possible autocorrelation [24]. To estimate the level and trend of the proportion of patients with a bleeding or thrombotic event before implementation of the multidisciplinary antithrombotic team, and to estimate the changes in level and trend after the implementation of the multidisciplinary antithrombotic team, the following linear regression model was used [25]:

$$\gamma_t = \text{ß}_0 + \text{ß}_1 * \text{time}_t + \text{ß}_2 * \text{intervention}_t + \text{ß}_3 * \text{time after intervention}_t + \mathbf{e}_t$$

$Y_0$ = mean percentage at time is $0 = \text{ß}_0$

$ß_1$ = baseline trend

$ß_2$ = immediate change after intervention

$ß_3$ = change in trend

For the secondary outcome and subgroup analyses regarding type of antithrombotic and hospital type logistic regression analysis was used, reporting odds ratio's (OR) and 95% confidence intervals (95% CI). In order to adjust for possible predictors, multivariable logistic regression analysis was performed. In-hospital, post discharge events and all-cause mortality were analysed separately by using logistic regression analysis. A t-test was used to determine the difference in the mean length of hospitalisation between both measurement periods. Costs were compared univariably using a t-test with bootstrapping (x1000).

Subgroup analyses were performed on the proportion of patients with a composite end point consisting of one or more bleeding or one or more thrombotic event from hospitalization until three months after hospitalisation stratified for each type of antithrombotic treatment (VKA, DOAC and LMWH) and for hospital type (Reinier de Graaf Hospital and the Erasmus University Medical Center).

Since the costs of non-major bleeding from earlier studies were based on hospitalised patients, a sensitivity analysis was performed. For patients with a non-major bleeding without hospitalisation three months after the initial hospital admission, costs for non-major bleeding were calculated by deducting the costs of hospitalisation.

## Results

### Study population

For final analysis 1,886 patients were included of which 941 in the usual care period, 469 in the RdGG and 472 in the EMC, and 945 in the intervention period, of which 473 in the RdGG and 472 in the EMC (Fig 1).

Characteristics of the included patients are presented in Table 1. The majority in both groups was male and the median age was 69 years. The two groups did not differ in gender, age, prior thrombotic event, hospital type, bodyweight, renal function, number of readmissions within three months after discharge and the number of patients who underwent surgery.

Patients included in the intervention period had significantly more previous bleeding events (28.5%) compared to patients in the usual care period (21%). The use of VKAs (58.4%) and LMWHs (44.8%) was significantly less in patients in the intervention period, compared to patients in the usual care period (VKAs, 68.8% and LMWHs, 51.9%). On the other hand, DOACs were used significantly more in the intervention period (27.8%), compared to patients in the usual care period (8.5%). The underlying diseases that indicated anticoagulant therapy were venous thromboembolism (48.2%), atrial fibrillation (42.9%), cardiac valve surgery (3.2%) and other reasons (5.7%).

### Effect of antithrombotic stewardship

Fig 2 shows the bleeding and thrombotic events during the study period. The segmented regression analysis showed that in the baseline period, i.e., before the introduction (baseline period) of the multidisciplinary team, the proportion of patients with a bleeding or thrombotic event at time = 0 was 9.49% (5.36 to 13.61) that showed an increase of 0.75% per 2 months (0.23% to 1.28%) during the baseline period. The introduction of the multidisciplinary team had no immediate impact on the event rate; the immediate effect was +1.63% (-3.60% to +6.85%). The slope of patients with a bleeding or thrombotic event after the introduction decreased significantly with -1.83% (-2.58% to -1.08%) per 2 months. Negative autocorrelation was detected (Durbin-Watson value of 3.51). In a sensitivity analysis we used a lag function to

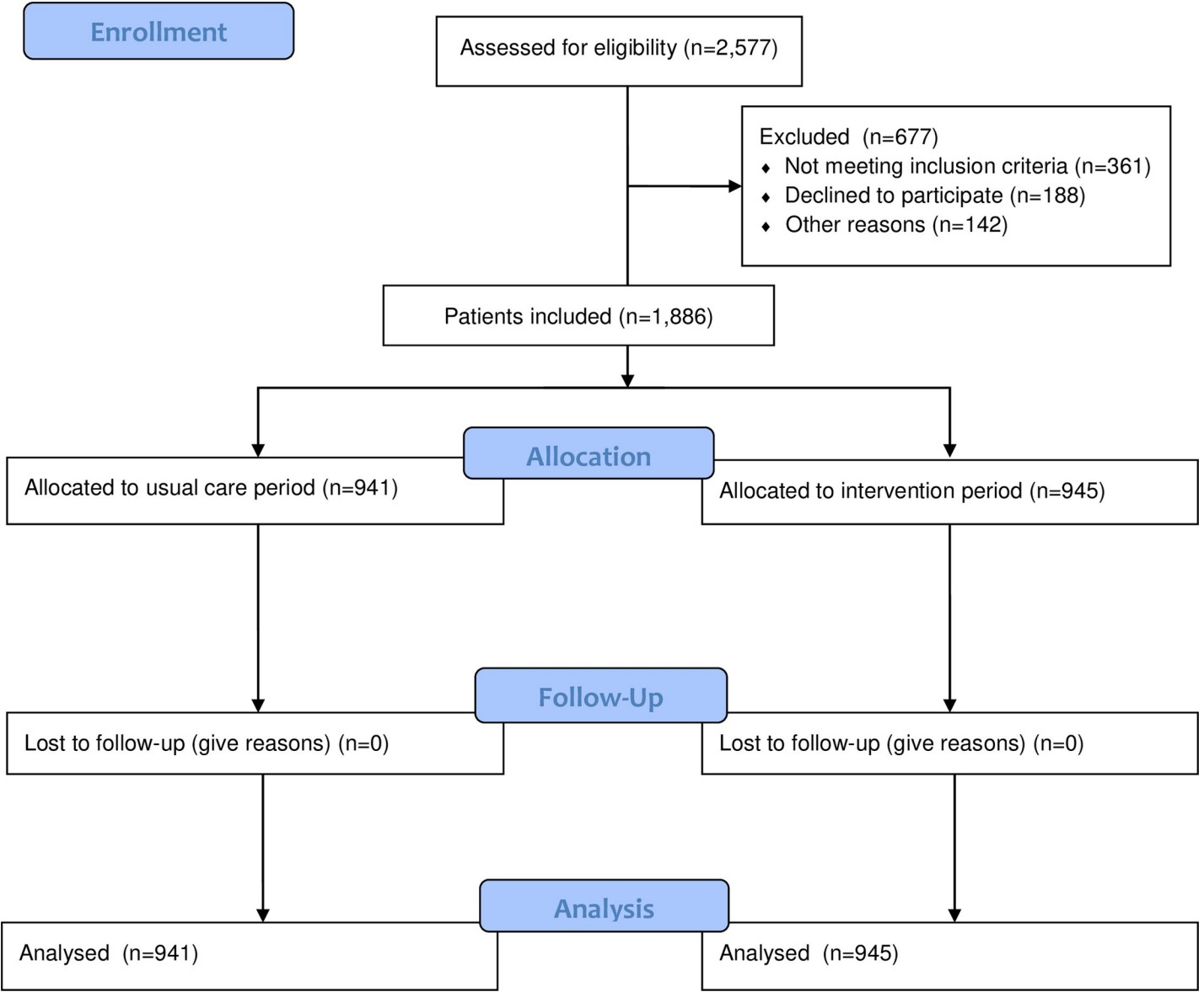

**Fig 1. Study flow.**

address the first order autocorrelation that resulted in no significant changes in the estimates of level and trend after the introduction of the multidisciplinary antithrombotic team.

## Secondary outcomes

Implementation of the multidisciplinary antithrombotic team showed no significant effect on the proportion of patients with a major bleeding event between the usual care period and intervention period (odds ratio [OR] 0.77; 95% confidence interval [95% CI] 0.55–1.10). The same applies to the proportion of patients with non-major bleeding events (OR 1.40; 95% CI 0.90–2.10) before and after implementation of the multidisciplinary antithrombotic team. Due to the low number of fatal and non-fatal thrombotic events before and after introduction of the multidisciplinary antithrombotic team no analysis was performed on the proportion of

**Table 1. Baseline characteristics of the patients.**

| Characteristic | Usual care period (*n* = 941) | Intervention period (*n* = 945) | p-value |
|---|---|---|---|
| Male gender | 562 (59.7) | 578 (61.2) | 0.522[$] |
| Age, years | 69 [59–77] | 69 [59–77] | 0.665[#] |
| Bleeding in history | 198 (21.0) | 269 (28.5) | **< 0.001**[$] |
| Thrombotic event in history | 448 (47.6) | 461 (48.8) | 0.610[$] |
| Hospital type, University Medical Center | 472 (50.2) | 472 (49.4) | 0.927[$] |
| Bodyweight, kg | 80 [70–91] | 80 [70–93] | 0.177[#] |
| e-GFR, $\leq$50 ml/min/1.73m$^2$ | 301 (33.0) | 266 (30.1) | 0.189[$] |
| Readmission within 3 months after discharge | 294 (31.2) | 291 (30.8) | 0.833[$] |
| Surgery | 340 (36.1) | 330 (34.9) | 0.583[$] |
| Type of anticoagulant therapy* | | | |
| - Vitamin K antagonist | 647 (68.8) | 552 (58.4) | **< 0.001**[$] |
| - Direct oral anticoagulant | 80 (8.5) | 263 (27.8) | **< 0.001**[$] |
| - Low-molecular-weight-heparin | 488 (51.9) | 423 (44.8) | **0.002**[$] |

Figures in bold are statistically significant.

Results are presented as median [interquartile range] or as number of patients (%) for non-continues data. N, number of patients at risk; *e-GFR* estimated glomerular filtration rate.

*Patients can use multiple anticoagulants during hospitalization.

[$]Chi square test.

[#]Mann-Whitney U test.

patients with a fatal and non-fatal thrombotic event between the usual care period and intervention period. Additional information regarding the total number of bleeding and thrombotic events, the severity and location of the bleeding and thrombotic events before and after implementation of the multidisciplinary antithrombotic team is listed in S2 Table.

## In-hospital and post discharge bleeding and thrombotic events

Implementation of the multidisciplinary antithrombotic team showed no significant effect on the proportion of patients with a composite end point consisting of one or more bleeding episode or one or more thrombotic event during hospitalization (OR 0.88 (0.62 to 1.24)) and the proportion of patients with a composite end point consisting of one or more bleeding or one or more thrombotic event after hospitalization (OR 1.00 (0.70 to 1.42)). After adjustment for characteristics differing between usual care and intervention period (i.e. bleeding in history and treatment with VKAs, DOACs or LMWHs) no significant differences were observed. Detailed data can be found in S3 Table.

## Subgroup analyses

Table 2 shows the proportion of patients with a composite end point consisting of one or more bleeding or one or more thrombotic event from hospitalization until three months after hospitalization per type of antithrombotic treatment and per hospital type. Logistic regression analysis revealed no significant differences in endpoint between the usual care period and intervention period for each type of antithrombotic treatment. An identical analysis has been performed for the type of hospital. In both the EMC and the RdGG no significant differences were found in the proportion of patients with the composite end point from hospitalization until 3 months after hospitalization between the usual care period and intervention period. Moreover, after adjustment for characteristics differing between usual care and intervention

| | $Y_0$ (95% CI) (mean percentage at time=0) | $\beta_1$ (95% CI) (baseline trend) | $\beta_2$ (95% CI) (immediate change) | $\beta_3$ (95% CI) (change in trend) |
|---|---|---|---|---|
| Bleeding and thrombotic events | **9.49** **(5.36 to 13.61)** | **0.75** **(0.23 to 1.28)** | 1.63 (-3.60 to 6.85) | **-1.83** **(-2.58 to -1.08)** |

Significant values are in bold type face

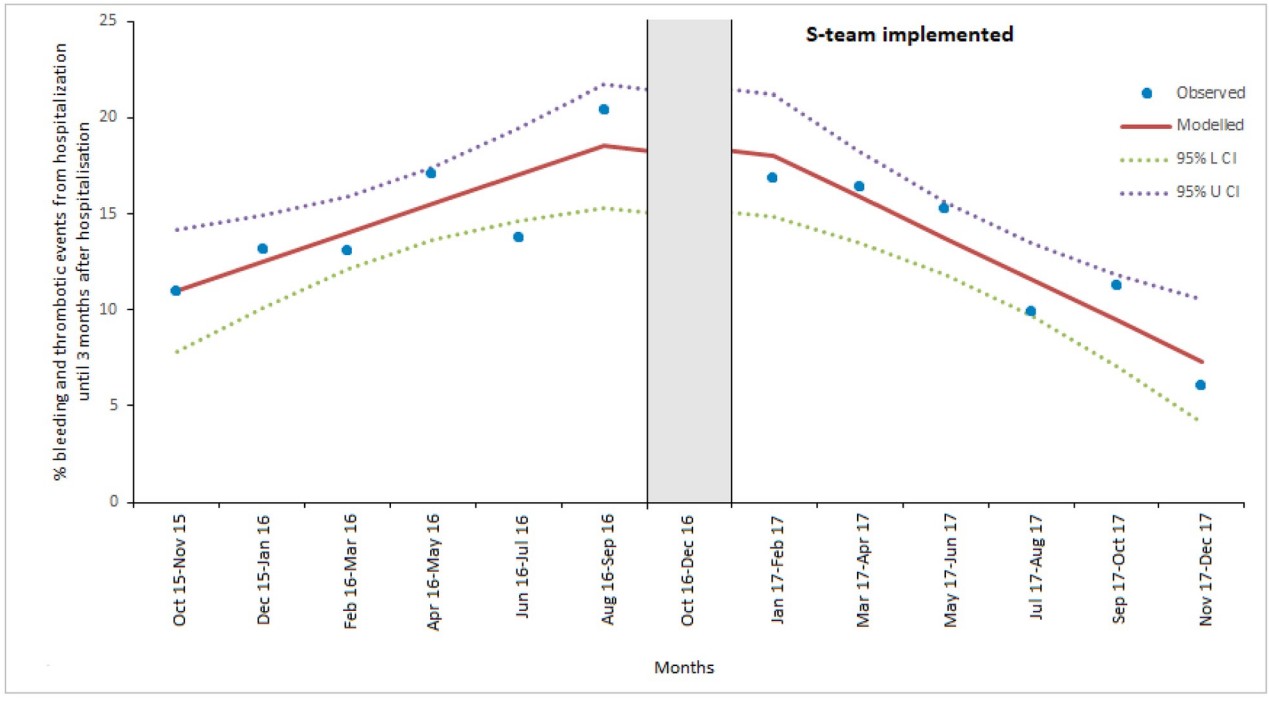

**Fig 2. Impact of antithrombotic stewardship (S-team) on bleeding and thrombotic events.** Vertical gray bar indicates the period in which the implementation of the multidisciplinary antithrombotic team took place.

**Table 2. Proportion of patients with a composite primary end point from hospitalization until 3 months after hospitalization stratified for each type of antithrombotic treatment and per type of hospital.**

| | Usual care period (*n* = 941) | Intervention period (*n* = 945) | | |
|---|---|---|---|---|
| | Bleeding and thrombotic events | Bleeding and thrombotic events | OR (95% CI) | ORadj [95% CI] |
| Type of anticoagulant therapy* | | | | |
| Vitamin K antagonist | 96/647 (14.8%) | 74/552 (13.4%) | 0.90 (0.64 to 1.23) | 0.85 (0.61 to 1.20) |
| Direct oral anticoagulant | 8/80 (10.0%) | 33/263 (12.5%) | 1.29 (0.57 to 2.92) | 1.10 (0.50 to 2.52) |
| Low-molecular-weight-heparin | 81/488 (16.6%) | 74/423 (17.5%) | 1.10 (0.75 to 1.51) | 0.99 (0.69 to 1.42) |
| Total | 185/1215 (15.2%) | 181/1238 (14.6%) | 0.96 (0.79 to 1.16) | 0.95 (0.90 to 1.01) |
| Hospital type | | | | |
| Reinier de Graaf Hospital | 53/469 (11.3%) | 47/473 (9.9%) | 0.87 (0.57 to 1.31) | 0.83 (0.54 to 1.29) |
| Erasmus University Medical Center | 82/472 (17.4%) | 77/472 (16.3%) | 0.93 (0.66 to 1.30) | 0.91 (0.63 to 1.30) |
| Total | 135/941 (14.3%) | 124/945 13.1% | 0.91 (0.73 to 1.15) | 0.95 (0.90 to 1.01) |

*OR* odds ratio, *95% CI* 95% confidence interval; ORadj, adjusted for characteristics differing between usual care and intervention period (bleeding in history and treatment with VKAs, DOACs or LMWHs).

*Patients can use multiple anticoagulants during hospitalization.

period (i.e. bleeding in history and treatment with VKAs, DOACs or LMWHs) no significant differences were observed.

## All-cause mortality and length of hospitalization

All-cause mortality was significantly lower in the intervention period [8.6% (81/945)] compared to the usual care period [11.5% (108/941)] OR 0.72 (0.53 to 0.98). Detailed data regarding the cause of death listed in S4 Table shows that the difference in all-cause mortality was not caused by death due to bleeding or thrombotic complications. The mean length of hospital stay was not significantly different with 11.8 days during the usual care period (standard deviation 13.7) versus 10.7 days (standard deviation 12.5) in the intervention period (p = 0.08).

## Economic evaluation of antithrombotic stewardship

Table 3 shows the costs of the implementation of antithrombotic stewardship during the usual care period and the intervention period of the study for each hospital separately.

## Erasmus university medical center

Mean S-team labour costs per admission were €44.80. The mean costs per admission for hospitalization days were €9360 in the usual care period (mean length of hospital stay was 14.58 days) and €8580 in the intervention period (mean length of hospital stay was 13.37 days). The number of bleeding events in the usual care period was 83; 54 major and 29 non-major bleeding events. The number of bleeding events in the intervention period was 82; 46 major and 36 non-major bleeding events. Multiplying the number of major and non-major bleeding events by the costs of the specific event resulted in €944 per admission for bleeding in the usual care period and in €908 per admission for bleeding in the intervention period. The number of thrombotic events in the usual care period was 16; 8 arterial thrombosis, 6 deep vein thrombosis and 2 pulmonary embolism. The number of thrombotic events in the intervention period was 14; 7 arterial thrombosis, 6 deep vein thrombosis and 1 pulmonary embolism. Multiplying the number of thrombotic events by the costs of the specific event resulted in €169 per admission for thrombotic events in the usual care period and in €143 per admission for thrombotic events in the intervention period. The total costs per admission of anticoagulant users decreased by €790 (£685; $894), but this was not statistically significant (p = 0.27).

## Reinier de Graaf Hospital

Mean S-team labour costs per admitted patient were €35.50. The mean costs per admission for hospitalization days were €3970 in the usual care period (mean length of hospital stay was 8.96 days) and €3570 in the intervention period (mean length of hospital stay was 8.06 days).

The number of bleeding events in the usual care period was 52; 35 major and 17 non-major bleeding events. The number of bleeding events in the intervention period was 48; 22 major and 26 non-major bleeding events, which resulted in €600 per admission for bleeding in the usual care period and in €514 per admission for bleeding in the intervention period. The number of thrombotic events in the usual care period was 9; 4 arterial thrombosis, 2 deep vein thrombosis and 3 pulmonary embolism, which resulted in €109 per admission for thrombotic events in the usual care period and in €77 per admission for thrombotic events in the intervention period. The total costs per admission of anticoagulant users decreased by €480 (£416; $544), but this was not statistically significant (p = 0.09).

**Table 3. Mean costs of usual care and of intervention per admission.**

| | Usual care period Mean costs per admission (€) | Intervention period Mean costs per admission (€) |
|---|---|---|
| **Erasmus Medical Center** | | |
| Labour costs S-team | - | € 44.80 (0–0) |
| S-team meetings[#] | - | € 0.30 |
| Medication reviews | - | € 32 |
| Patient empowerment | - | € 12 |
| Maintenance of anticoagulant therapy protocols and education | - | € 0.50 |
| Costs for hospitalization days | € 9360 (3852–10,914) | € 8580 (2729–10,914) |
| Costs for bleeding | € 944 (0–0) | € 908 (0–0) |
| Non-major bleeding | € 267 | € 332 |
| Major bleeding | € 677 | € 576 |
| Costs for thrombotic events | € 169 (0–0) | € 143 (0–0) |
| Arterial thrombosis | € 81 | € 71 |
| Deep vein thrombosis | € 56 | € 56 |
| Pulmonary embolism | € 32 | € 16 |
| **Total costs (p = 0.27)** | **€ 10470 (3852–13,482)** | **€ 9680 (3254–12,766)** |
| **Reinier de Graaf Hospital** | | |
| Labour costs S-team | - | € 35.50 (0–0) |
| S-team meetings[#] | - | € 0.50 |
| Medication reviews | - | € 23 |
| Patient empowerment | - | € 10 |
| Maintenance of anticoagulant therapy protocols and education | - | € 2 |
| Costs for hospitalization days | € 3970 (1772–4873) | € 3570 (1329–4430) |
| Costs for bleeding | € 600 (0–0) | € 514 (0–0) |
| Non-major bleeding | € 158 | € 239 |
| Major bleeding | € 442 | € 275 |
| Costs for thrombotic events | € 109 (0–0) | € 77 (0–0) |
| Arterial thrombosis | € 41 | € 51 |
| Deep vein thrombosis | € 19 | € 10 |
| Pulmonary embolism | € 49 | € 16 |
| **Total costs (p = 0.09)** | **€ 4680 (1772–6202)** | **€ 4200 (1364–5547)** |

[#]Calculated on the total number of hospitalized patients treated with therapeutic anticoagulant medication per year.

*S-team* antithrombotic stewardship.

Results are presented as mean costs (interquartile range).

The sensitivity analysis for costs of non-major bleeding in which a distinction is made between admitted and non-admitted patients with a non-major bleeding three months after hospitalization, showed similar total costs in the usual and intervention period (S5 Table).

## Discussion

The S-team study shows that implementation of a multidisciplinary antithrombotic team over time significantly reduces the composite end point consisting of one or more bleeding episodes or one or more thrombotic event from hospitalization until three months after hospitalization in patients using anticoagulant drugs. Additionally, implementation of a multidisciplinary

antithrombotic team appeared to result in lower all-cause mortality. The present study showed no significant effect of the intervention on the severity of bleeding, in-hospital events, post-discharge events, mean length of hospital stay and costs. Moreover, no significant effect of the intervention was found for type of antithrombotic treatment and hospital type.

Multifaceted intervention studies have been shown to improve the safety of antithrombotic therapy. These studies focused mainly on surrogate endpoints such as compliance to antithrombotic protocols, patient care and transitioning of patients on anticoagulation to outpatient management [7,26,27]. Most studies used a pre-post analysis to determine the impact of an anticoagulant stewardship program [7,26]. In contrast to previous studies, our interrupted time series study design with segmented linear regression analysis is more robust and clinically more relevant since it evaluates the longitudinal effect of the implementation of a hospital-based antithrombotic stewardship and adjusts for trends [25]. Therefore, our study design compares favourably to other studies.

We found a significant upward trend in the proportion of patients with the primary endpoint in the usual care period. At the time of the usual care period DOACs were introduced in both hospitals. The lack of experience among clinicians with these relatively new drugs in daily practice may have contributed to inappropriate use leading to an increase in bleeding and thrombotic complications. However, as DOACs have limited drug interactions, they do not require coagulation monitoring. They are easier to use than VKA and do not require extensive experience. Furthermore, large randomized controlled clinical trials have shown DOACs to be at least as effective as VKAs, and have found them to be associated with a lower or comparable risk of bleeding complications [28–32]. For this reason, the introduction of DOACs does not seem not to be the explanation for the upward trend. Possibly other unknown reasons are responsible for the increase in the proportion of patients with the primary endpoint in the usual care period.

In our study, implementation of the multidisciplinary antithrombotic team appeared to result in lower all-cause mortality, especially in patients with cancer, kidney diseases and respiratory diseases. Lower all-cause mortality is in line with a study performed by Bond et al [33]. They found that pharmacist-managed heparin and warfarin therapy had a profound effect on improving healthcare outcomes in Medicare patients. In hospitals without pharmacist-provided heparin management, death rates were 11.41% higher and in hospitals without pharmacist-provided warfarin management, death rates were 6.20% higher.

Our economic evaluation showed that implementation of a multidisciplinary antithrombotic team was accompanied by a reduction in the total costs in both hospitals. Costs per admission of anticoagulant users decreased by €790 (£685; $894) in the university medical center and €480 (£416; $544) in the general hospital, but this was not statistically significant. This finding is in line with the antithrombotic stewardship program of Padron et al. who managed to save $661 per patient [7].

## Strengths and limitations

This study is the first study on the effect of hospital-based multidisciplinary antithrombotic stewardship on the clinically relevant primary endpoint composed of bleedings and thrombotic events. Furthermore, cost-effectiveness of our multi-interventional strategy was included. Our study was performed in two different types of hospitals, a university medical centre and a large general hospital. This increases the generalizability of our findings.

The study was designed in a way that minimizes bias in the primary outcome, by using the generally accepted objective ISTH criteria and by blinding all case record forms for the adjudication of the endpoint bleeding and thrombotic events with respect to the study period by the

two independent expert physicians. Segmented regression analysis for the interrupted time series was used for analysis of the primary outcome, making it possible to evaluate the longitudinal effect of the intervention and to adjust for trends. An interrupted time series design is more valid than a simple before-after design, that is commonly used in studies on complex health policy interventions where a control group is difficult.

Our study has several limitations. Data on bleeding and thrombotic events occurring during hospitalization were derived from reports of the responsible physicians in the electronic medical records (EMRs). Post discharge data were from reports of the patient's general practitioner and/or the patient himself. This makes the study dependent on the information recorded by the responsible physician, general practitioner or patient, which may lead to underreporting, especially of the non-major bleeding events in the usual care period without additional therapeutic education. Logistic regression analysis was used for analysis of the secondary outcome all-cause mortality, making the effects of time and trends are missing. Therefore, the effect of the multidisciplinary antithrombotic team on all-cause mortality should be interpreted with caution. Finally, our intervention is multifaceted making it impossible to know which specific intervention (e.g. medication reviews) had the largest influence on the effect and safety of antithrombotic therapy during and after hospitalization.

## Conclusions

Implementation of a multidisciplinary antithrombotic team was associated with a reduction in the proportion of patients with complications associated with the use of anticoagulant drugs. Furthermore, lower all-cause mortality was observed. The significant downward trend after implementation of a multidisciplinary antithrombotic team continues after the last data point in the intervention period which may indicate that the effect of the multidisciplinary antithrombotic team on the proportion of patients with bleeding and thrombotic events and on mortality may have been even larger if the study had a longer follow-up. Therefore additional research on the long term effects of the intervention would be of interest. Furthermore, future research should focus on which intervention(s) of the multifaceted approach had the most influence on the outcomes and which patients are at the highest risk and would benefit the most from implementation of the multidisciplinary antithrombotic team.

## Supporting information

**S1 Checklist. TREND statement checklist.**
(PDF)

**S1 Table. Data collection.** *e-GFR* estimated glomerular filtration rate, *INR* International Normalized Ratio.
(PDF)

**S2 Table. Characterization of all bleeding and thrombotic events before and after implementation of the multidisciplinary antithrombotic team.** These bleeding and thrombotic events occurred in 135 patients in the usual care period and 124 patients in the intervention period.
(PDF)

**S3 Table. Proportion of patients with a composite end point consisting of ≥1 bleeding or ≥1 thrombotic event during and 3 months after hospitalization.** *OR* odds ratio, *95% CI* 95% confidence interval.
(PDF)

**S4 Table. Causes of death.**
(PDF)

**S5 Table. Sensitivity analysis for costs of non-major bleeding.**
(PDF)

**S1 File.**
(PDF)

**S2 File.**
(PDF)

**S3 File.**
(DOCX)

## Acknowledgments

We would like to express our gratitude to all professionals who are part of the multidisciplinary antithrombotic teams and to the hospital pharmacists in the Erasmus University Medical Center and the Reinier de Graaf Hospital for their participation in performing medication reviews. Our special thanks go to Marleen de Graaf-van der Kort (thrombotic nurse Reinier de Graaf Hospital), Wilma van Dongen (cardiology nurse Reinier de Graaf Hospital), Mariska Gieling (cardiology nurse Reinier de Graaf Hospital), Eugene Pruissen (cardiology nurse Reinier de Graaf Hospital), Wilma Neeleman-de Zeeuw (thrombotic nurse Erasmus University Medical Center) and the research students (Sebnem Aybike Akgöl, Lamyae Maanach, Jonathan Knikman, Vera Bukkems, Jennifer Hollander, Krishnika Jeyasimman, Shamayel Mobayyen, Halat Naby, Pawan Rauf) for their participation in the patient inclusion and collection of patient data.

## Author Contributions

**Conceptualization:** Albert R. Dreijer, Marieke J. H. A. Kruip, Jeroen Diepstraten, Frank W. G. Leebeek, Patricia M. L. A. van den Bemt.

**Data curation:** Albert R. Dreijer.

**Formal analysis:** Albert R. Dreijer, Suzanne Polinder, Peter G. M. Mol, Patricia M. L. A. van den Bemt.

**Funding acquisition:** Albert R. Dreijer.

**Investigation:** Albert R. Dreijer.

**Methodology:** Albert R. Dreijer, Suzanne Polinder, Peter G. M. Mol, Frank W. G. Leebeek, Patricia M. L. A. van den Bemt.

**Project administration:** Albert R. Dreijer.

**Resources:** Albert R. Dreijer, Patricia M. L. A. van den Bemt.

**Software:** Albert R. Dreijer.

**Supervision:** Marieke J. H. A. Kruip, Jeroen Diepstraten, Suzanne Polinder, Frank W. G. Leebeek, Patricia M. L. A. van den Bemt.

**Validation:** Albert R. Dreijer.

**Visualization:** Albert R. Dreijer.

Writing – **original draft:** Albert R. Dreijer.

Writing – **review & editing:** Albert R. Dreijer, Marieke J. H. A. Kruip, Jeroen Diepstraten, Suzanne Polinder, Rolf Brouwer, Peter G. M. Mol, F. Nanne Croles, Esther Kragten, Frank W. G. Leebeek, Patricia M. L. A. van den Bemt.

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
