## [Decision Letter · Decision Letter 0]

20 Jan 2020

PONE-D-19-31280

Effect of antithrombotic stewardship on the efficacy and safety of antithrombotic therapy during and after hospitalization

PLOS ONE

Dear Dr Dreijer,

Thank you for submitting your manuscript to PLOS ONE. After careful consideration, we feel that it has merit but does not fully meet PLOS ONE’s publication criteria as it currently stands. Therefore, we invite you to submit a revised version of the manuscript that addresses the comments you will find below.

We would appreciate receiving your revised manuscript by Mar 05 2020 11:59PM. To enhance the reproducibility of your results, we recommend that if applicable you deposit your laboratory protocols in protocols.io, where a protocol can be assigned its own identifier (DOI) such that it can be cited independently in the future. For instructions see: http://journals.plos.org/plosone/s/submission-guidelines#loc-laboratory-protocols

We look forward to receiving your revised manuscript.

Kind regards,

Christophe Leroyer

Academic Editor

PLOS ONE

Journal Requirements:

2. We noticed you have some minor occurrence(s) of overlapping text with the following previous publication(s), which needs to be addressed:

https://doi.org/10.1007/s11096-019-00834-2

https://doi.org/10.1016/j.ejim.2019.01.008

In your revision ensure you cite all your sources (including your own works), and quote or rephrase any duplicated text outside the Methods section. Further consideration is dependent on these concerns being addressed.

Stichting Phoenix Schiedam, the pharmaceutical companies (Daiichi Sankyo, Boehringer Ingelheim, Bayer and Pfizer) and the Scientific Committee Reinier de Graaf Gasthuis provided financial support for this study in the form of unrestricted grants. The funders had no role in study design, data collection and analysis, decision to publish, or preparation of the manuscript.

We note that you received funding from a commercial source: Daiichi Sankyo, Boehringer Ingelheim, Bayer and Pfizer.

**Comments to the Author**

1. Is the manuscript technically sound, and do the data support the conclusions?

Reviewer #1: Partly

Reviewer #2: Yes

2. Has the statistical analysis been performed appropriately and rigorously? 

Reviewer #1: Yes

Reviewer #2: I Don't Know

3. Have the authors made all data underlying the findings in their manuscript fully available?

Reviewer #1: Yes

Reviewer #2: Yes

4. Is the manuscript presented in an intelligible fashion and written in standard English?

Reviewer #1: Yes

Reviewer #2: Yes

5. Review Comments to the Author

Reviewer #1: Interesting study with some technical merits.

However, there are major limits that should explained and discussed; additional statistics are required on adjustment analyses:

First: the author did not confirm their primary hypothesis of 9% composite before and 5.5% after intervention and there was no reduction at all based on this prevision. In addition, no multivariable analysis was done (ie, no adjustment on variables differently distributed between the usual and the intervention group): this should be done, because there were strong differences in proportion of DOACs. Was there a predefined statistical plan analysis ? Was the use of segmental regression analysis pre-planned?

Finally, the conclusion should be lowered: the study suggests a potential improvement, but there is no demonstration.

Second: the underlying disease that indicated anticoagulant therapy is not described; this is a major limitation since VTE and atrial fibrillation, for example, are not the same diseases at all and the thrombotic complications are different in terms of frequency and type of event.

Other:

- Table 2: a line with the total composite is needed in each period.

- In the flow chart, authors indicate randomization, which is wrong.

Reviewer #2: This is a truly relevant study. The evaluation is quite complete, including clinical and economic criteria. The implementation of a multidisciplinary team in charge of therapeutic education and medication review may prevent interactions, duplicate medications, and dosing errors. Therefore, we can assume it could reduce the thrombotic and bleeding events. I don’t see any major issue in the manuscript. You will find below a few minor suggestions.

Abstract

line 21: The primary endpoint could be described in the methods section.

Methods

line 171: Some other systemic embolisms could have been included. The decision to only include myocardial infraction and ischemic stroke as arterial thrombosis could be justified.

Discussion

line 373-379: Considering NOACs have limited drug interactions, do not require routine coagulation monitoring with fixed doses, they are easier to use than VKA. Their use does not require a extensive experience.

line 414: It might have been less occurrences of underreporting of the non major bleeding events in the S-Team group. Indeed we can assume that after a therapeutic education, patients might have been more inclined to notify a non major bleeding.

6. PLOS authors have the option to publish the peer review history of their article (what does this mean?). If published, this will include your full peer review and any attached files.

Reviewer #1: No

Reviewer #2: No

---

## [Author Response · Author response to Decision Letter 0]

28 Feb 2020

Our responses are listed in the response letter.

---

## [Decision Letter · Decision Letter 1]

19 May 2020

PONE-D-19-31280R1

Effect of antithrombotic stewardship on the efficacy and safety of antithrombotic therapy during and after hospitalization

PLOS ONE

Dear Dr Dreijer,

Please accept my apologies for this delay: our reviewers have been busy these previous days and thank you for resubmitting your manuscript to PLOS ONE. After careful consideration, we feel that it has merit but does not fully meet PLOS ONE’s publication criteria as it currently stands and thus few minor revisions are needed. Therefore, we invite you to submit a revised version of the manuscript that addresses the points raised during the review process and you will find below.

We would appreciate receiving your revised manuscript. To enhance the reproducibility of your results, we recommend that if applicable you deposit your laboratory protocols in protocols.io, where a protocol can be assigned its own identifier (DOI) such that it can be cited independently in the future. For instructions see: http://journals.plos.org/plosone/s/submission-guidelines#loc-laboratory-protocols

We look forward to receiving your revised manuscript.

Kind regards,

Christophe Leroyer

Academic Editor

PLOS ONE

**Comments to the Author**

1. If the authors have adequately addressed your comments raised in a previous round of review and you feel that this manuscript is now acceptable for publication, you may indicate that here to bypass the “Comments to the Author” section, enter your conflict of interest statement in the “Confidential to Editor” section, and submit your "Accept" recommendation.

Reviewer #1: All comments have been addressed

Reviewer #3: (No Response)

2. Is the manuscript technically sound, and do the data support the conclusions?

Reviewer #1: Yes

Reviewer #3: Yes

3. Has the statistical analysis been performed appropriately and rigorously? 

Reviewer #1: Yes

Reviewer #3: Yes

4. Have the authors made all data underlying the findings in their manuscript fully available?

Reviewer #1: Yes

Reviewer #3: Yes

5. Is the manuscript presented in an intelligible fashion and written in standard English?

Reviewer #1: Yes

Reviewer #3: Yes

6. Review Comments to the Author

Reviewer #1: all the questions have been addressed. I do not have further comments and the manuscript is suitable for publication

Reviewer #3: The primary aim, as stated, of the prospective nonrandomized clinical trial was to evaluate the efficacy and safety of antithrombotic therapy during and after hospitalization. A group of patients received the intervention protocol while another group served as controls. Although a significant change in the proportion of patients with a thrombotic event was not apparent immediately, by 2 months a significant decrease was observed.

Minor revisions:

1- Abstract: Consider beginning the following sentence with the phrase, The primary aim was to estimate the proportion of patients ... “Primary outcome was the proportion

of patients ... with a composite end point consisting of one or more bleeding episodes or

one or more thrombotic event from hospitalization until three months after hospitalization.”

2- Abstract: Provide more clarity in the following statement. Specifically identify the primary endpoint. “Introduction of the multidisciplinary team had no immediate impact +1.63% (-3.60% to +6.85%) on the primary endpoint, but over time the primary endpoint event rate decreased significantly with -1.83% (-2.58% to -1.08%) per 2 month period.”

3- Line 190: Indicate the statistical testing method which attains 80% power.

4- Line 232: Clarify the type of subgroup analysis performed.

5- Line 282: Clarify that the numbers in parentheses represent the confidence interval.

6- Table 3: Provide a measure of dispersion corresponding to each mean costs.

7- Provide a summary of adverse events to address the safety aim.

8- Identify the sources of funding for the study.

9- The primary aim of the study does not appear to be to determine efficacy and safety, but to compare the proportion of patients in the two groups who develop thrombotic events. Please clarify.

7. PLOS authors have the option to publish the peer review history of their article (what does this mean?). If published, this will include your full peer review and any attached files.

Reviewer #1: No

Reviewer #3: No

---

## [Author Response · Author response to Decision Letter 1]

5 Jun 2020

Reviewer comments of PLOS ONE PONE-D-19-31280R1

Effect of antithrombotic stewardship on the efficacy and safety of antithrombotic therapy during and after hospitalization

Reviewer #1

All the questions have been addressed. I do not have further comments and the manuscript is suitable for publication.

Reviewer #3

The primary aim, as stated, of the prospective nonrandomized clinical trial was to evaluate the efficacy and safety of antithrombotic therapy during and after hospitalization. A group of patients received the intervention protocol while another group served as controls. Although a significant change in the proportion of patients with a thrombotic event was not apparent immediately, by 2 months a significant decrease was observed.

Minor revisions:

1. Abstract: Consider beginning the following sentence with the phrase, The primary aim was to estimate the proportion of patients ... “Primary outcome was the proportion

of patients ... with a composite end point consisting of one or more bleeding episodes or

one or more thrombotic event from hospitalization until three months after hospitalization.”

We have adjusted the sentence in the Abstract section of the manuscript. Page 2, lines 19-21.

 “The primary aim was to estimate the proportion of patients with a composite end point consisting of one or more bleeding episodes or one or more thrombotic event from hospitalization until three months after hospitalization”. 

2. Abstract: Provide more clarity in the following statement. Specifically identify the primary endpoint. “Introduction of the multidisciplinary team had no immediate impact +1.63% (-3.60% to +6.85%) on the primary endpoint, but over time the primary endpoint event rate decreased significantly with -1.83% (-2.58% to -1.08%) per 2 month period.”

We have clarified the results in the Abstract section. Page 2 and 3, lines 28-32.

“The S-team study showed that implementation of a multidisciplinary antithrombotic team over time significantly reduced the composite end point consisting of one or more bleeding episodes or one or more thrombotic event from hospitalization until three months after hospitalization in patients using anticoagulant drugs (-1.83% (-2.58% to -1.08%) per 2 month period)”

3. Line 190: Indicate the statistical testing method which attains 80% power.

Because a sample size calculation was not possible for an interrupted time series analysis, we have used the Chi Square test on the advice of the statistician. We have indicated the statistical testing method in the sample size section of the manuscript. Page 10, lines 195-197.

“A Chi Square test with a type 1 error of 0.05, power 80% resulted in a sample size of 1,834 patients.”

4. Line 232: Clarify the type of subgroup analysis performed.

We have clarified the type of subgroup analyses. Page 12, lines 235-239.

“Subgroup analyses were performed on the proportion of patients with a composite end point consisting of one or more bleeding or one or more thrombotic event from hospitalization until three months after hospitalization stratified for each type of antithrombotic treatment (VKA, DOAC and LMWH) and for hospital type (Reinier de Graaf Gasthuis and the Erasmus University Medical Center).”

5. Line 282: Clarify that the numbers in parentheses represent the confidence interval.

We have clarified the numbers in parentheses representing the confidence intervals. Page 14, lines 284-288.

“Implementation of the multidisciplinary antithrombotic team showed no significant effect on the proportion of patients with a major bleeding event between the usual care period and intervention period (odds ratio [OR] 0.77; 95% confidence interval [95% CI] 0.55-1.10). The same applies to the proportion of patients with non-major bleeding events (OR 1.40; 95% CI 0.90-2.10) before and after implementation of the multidisciplinary antithrombotic team”

6. Table 3: Provide a measure of dispersion corresponding to each mean costs.

We have added the interquartile ranges (IQR) to the mean costs in Table 3 on page 25. 

7. Provide a summary of adverse events to address the safety aim.

Because bleeding and thrombotic events are the most important adverse events of antithrombotic therapy we have chosen to register only that type of adverse event. Characterization of all bleeding and thrombotic events before and after implementation of the multidisciplinary antithrombotic team are listed in Table S2. 

8. Identify the sources of funding for the study.

The sources of funding for this study are described on page 22, lines 488-493.

“Stichting Phoenix Schiedam, the pharmaceutical companies (Daiichi Sankyo, Boehringer Ingelheim, Bayer and Pfizer) and the Scientific Committee Reinier de Graaf Gasthuis provided financial support for this study in the form of unrestricted grants. The funders had no role in study design, data collection and analysis, decision to publish, or preparation of the manuscript and this does not alter our adherence to PLOS ONE policies on sharing data and materials.”

9. The primary aim of the study does not appear to be to determine efficacy and safety, but to compare the proportion of patients in the two groups who develop thrombotic events. Please clarify.

We have adjusted the primary aim in the background section of the abstract. Page 2, lines 12-15.

“The primary aim of this study was to compare the proportion of patients with a composite end point consisting of one or more bleeding episodes or one or more thrombotic event from hospitalization until three months after hospitalization.”

---

## [Editor Report · Decision Letter 2]

9 Jun 2020

Effect of antithrombotic stewardship on the efficacy and safety of antithrombotic therapy during and after hospitalization

PONE-D-19-31280R2

Dear Dr. Dreijer,

We’re pleased to inform you that your manuscript has been judged scientifically suitable for publication and will be formally accepted for publication once it meets all outstanding technical requirements.

Kind regards,

Christophe Leroyer

Academic Editor

PLOS ONE

---

## [Editor Report · Acceptance letter]

12 Jun 2020

PONE-D-19-31280R2 

Effect of antithrombotic stewardship on the efficacy and safety of antithrombotic therapy during and after hospitalization 

Dear Dr. Dreijer:

I'm pleased to inform you that your manuscript has been deemed suitable for publication in PLOS ONE. Congratulations! Your manuscript is now with our production department. 

Kind regards, 

on behalf of

Dr. Christophe Leroyer 

Academic Editor

PLOS ONE